# In Vivo Rapid Investigation of CRISPR-Based Base Editing Components in *Escherichia coli* (IRI-CCE): A Platform for Evaluating Base Editing Tools and Their Components

**DOI:** 10.3390/ijms23031145

**Published:** 2022-01-20

**Authors:** Rahul Mahadev Shelake, Dibyajyoti Pramanik, Jae-Yean Kim

**Affiliations:** 1Division of Applied Life Science (BK21 FOUR Program), Plant Molecular Biology and Biotechnology Research Center, Gyeongsang National University, Jinju 52828, Korea; dpinbiotech@gmail.com; 2Division of Life Science, Gyeongsang National University, 501 Jinju-daero, Jinju 52828, Korea

**Keywords:** base editing, CRISPR, genome editing, heterologous expression

## Abstract

Rapid assessment of clustered regularly interspaced short palindromic repeats/CRISPR-associated protein (CRISPR/Cas)-based genome editing (GE) tools and their components is a critical aspect for successful GE applications in different organisms. In many bacteria, double-strand breaks (DSBs) generated by CRISPR/Cas tool generally cause cell death due to the lack of an efficient nonhomologous end-joining pathway and restricts its use. CRISPR-based DSB-free base editors (BEs) have been applied for precise nucleotide (nt) editing in bacteria, which does not need to make DSBs. However, optimization of newer BE tools in bacteria is challenging owing to the toxic effects of BE reagents expressed using strong promoters. Improved variants of two main BEs, cytidine base editor (CBE) and adenine base editor (ABE), capable of converting C to T and A to G, respectively, have been recently developed but yet to be tested for editing characteristics in bacteria. Here, we report a platform for in vivo rapid investigation of CRISPR-BE components in *Escherichia coli* (IRI-CCE) comprising a combination of promoters and terminators enabling the expression of nCas9-based BE and sgRNA to nontoxic levels, eventually leading to successful base editing. We demonstrate the use of IRI-CCE to characterize different variants of CBEs (PmCDA1, evoCDA1, APOBEC3A) and ABEs (ABE8e, ABE9e) for bacteria, exhibiting that each independent BE has its specific editing pattern for a given target site depending on protospacer length. In summary, CRISPR-BE components expressed without lethal effects on cell survival in the IRI-CCE allow an analysis of various BE tools, including cloned biopart modules and sgRNAs.

## 1. Introduction

The adoption of the bacterial immune system, clustered regularly interspaced short palindromic repeats/CRISPR-associated protein (CRISPR/Cas), for targeted genetic manipulation has transformed the field of genome editing (GE) research [1]. The CRISPR-based tools allow direct or indirect programmable engineering of all three major building blocks of life, i.e., DNA, RNA, and protein [2]. In eukaryotes, double-strand breaks (DSBs) generated by CRISPR/Cas tool are repaired by error-prone nonhomologous end-joining (NHEJ) pathway leading to the gene disruption. Most bacteria lack the efficient NHEJ pathway, thus restricting the DSB-dependent CRISPR/Cas applications [3]. The DSB-free DNA base editors (BEs) are emerging tools to install precise mutations at target regions in bacteria [4].

In BE systems, deaminase fusion with Cas enzyme allows precise base conversion in the targeted region. The BE tools are based on the use of nickase (nCas9) or dead Cas9 (dCas9) (mainly from *Streptococcus pyogenes*) that does not need to make DSBs in the target DNA [5,6]. In animals and plants, primarily nCas9-BE is employed to stimulate the nicking of non-deaminated strands that produces single-strand breaks (SSBs) and favor desired BE outcomes by tweaking the cellular DNA damage responses [7]. The single guide RNA (sgRNA) together with the nCas9-BE or dCas9-BE complex binds the protospacer adjacent motif (PAM) in the target locus to form an R-loop, in which the partially open nontarget strand (NTS) provides a possible substrate for base conversion. Depending on the type of deaminase used, editing can occur within or near the protospacer, described as deamination or editing window [8].

Base editing tools have recently been optimized in various microbes due to their simplicity, high efficiency, and precision. However, toxic effects of BE components (deaminase, nCas9(D10A), uracil-DNA glycosylase inhibitor (UGI) protein) were earlier reported in several bacterial species (Table 1). For example, expression of some components of CRISPR-BE machinery shown poor transformation efficiency, such as nCas9(D10A), nCas9(D10A)-PmCDA, and dCas9-PmCDA1-1×UGI in *Escherichia coli* [9]; rAPOBEC1-nCas9(D10A) in *Pseudomonas* sp. [10]; nCas9(D10A)-PmCDA1 in *Bacillus subtilis* [11,12]; and dCas9-PmCDA1-1×UGI-LVA.Tag in *Streptomyces* spp. [13] and *Agrobacterium* sp. [14]. Higher expression of BE reagents was reported to adversely affect cell growth and BE efficiency [12]. The use of an optimal promoter facilitated successful base editing in *Bacillus* sp. [12], indicating an appropriate amount of BE expression as a crucial factor in achieving higher BE efficiency. Nevertheless, using an inducible promoter [10] or optimal constitutive promoter [15] circumvents the toxic effects of BE components. Previously, inducible promoter-mediated leaky expression (even without adding the inducer chemicals) was sufficient to achieve efficient base editing [11,12,13,14,16]. In this regard, we propose finding the promoters expressing the optimal amount of BE components, including sgRNAs that may help avoid cell toxicity, and analyzing BE tools. The use of validated CRISPR-BE biopart modules in Golden Gate assembly protocols [17] might be helpful for both prokaryotic and plant BE experiments.

Here, a BE system was employed to establish an efficient and rapid evaluation platform to assess the differential-strength promoter-driven base conversion in *E. coli*, termed in vivo rapid investigation of CRISPR-BE components in *E. coli* (IRI-CCE). We report for the first time that the efficient sgRNA expression was driven by the consensus sequence of Arabidopsis U6 promoter (pAtU6) (75 bp) in *E. coli* cells. The modular cloning approach described in the present work enables the validation of designed bioparts for further modular cloning of CRISPR-BE plasmids and the interchangeable use of bioparts independent of its context in the IRI-CCE platform. Different versions of cytidine base editor (CBE) and adenine base editor (ABE) were analyzed for editing features and substitution efficiency from cytosine to thymine (C to T) and adenine to guanine (A to G), respectively. Editing characteristics of BEs, including CBEs (Target-AID system based on sea lamprey cytidine deaminase, PmCDA1 [6]; evolved variant of PmCDA1, evoCDA1 [18]; human APOBEC3A, A3A [19] and ABEs (ABE8e [20] and ABE9e [21]) were optimized. This is the first report of optimization of evoCDA1, APOBEC3A, ABE8e, ABE9e, and pAtU6-mediated gRNA expression for BE studies in *E. coli*. Moreover, editing outcomes of the analyzed set of spacers (gRNAs) of different lengths imply that the gRNA length is crucial in achieving variable sizes of editing windows.

## 2. Result

### 2.1. Optimizing Base Editing Platform Based on PmCDA1-Mediated C-to-T Conversion in E. coli

The CRISPR-based BE construct consists of two major transcriptional units: deaminase-fused Cas9 mutant and sgRNA expression unit, driven by two separate promoters. Previous bacterial BE studies consisted of a deaminase fusion with partially impaired nCas9 or inactive dCas9 [9,14,22,23]. We chose the nCas9(D10A)-based CBE system in the present study because of the frequent use in BE studies in different organisms. Partially active nCas9 generates the SSBs that favor desired BE outcomes [7].

#### 2.1.1. Selecting Nontoxic Promoters for BE Studies in *E. Coli*

As observed in earlier studies, the higher expression of BE components adversely affects cell survival in bacteria [9,14]. To avoid the lethal effect of nCas9-BE (examples of cell toxicity also summarized in Table 1), we opted to explore different promoters for the expression of CRISPR-BE machinery without affecting cell survival. Among the tested promoters, we observed poor transformation efficiency with a mix of incorrect clones in the case of high-strength bacterial promoter pEc1 for CBE testing (Appendix A). When CBE (nCas9(D10A)-PmCDA1-1×UGI or dCas9(D10A+H840A)-PmCDA1-1UGI) was expressed by pEc1 together with pJ23119-sgRNA, no correct clones were obtained, possibly due to the toxicity caused by BE components in *E. coli* (Appendix A).

Bacterial pGlpT (BBa_J72163) promoter [24,25] showed optimal expression activity but less activity than pEc1 in transcript analysis of superfolder green fluorescent protein (sfGFP) (Appendix A). Additionally, cauliflower mosaic virus (CaMV) 35S promoter (p35S) was selected to evaluate the expression activity of nCas9-BE components in *E. coli*. Although the p35S is less active in *E. coli* [26,27], this aspect is of great significance for rapidly validating cloned plasmids in *E. coli* or designing a heterologous expression platform consisting of differential-strength promoters (Figure 1A). The use of validated CRISPR-BE biopart modules in Golden Gate assembly protocols might be helpful for BE experiments allowing compatible use in both bacteria and plants. In several instances, cloning new plasmid vectors, genetic delivery of CRISPR plasmids, and analysis of GE outcomes are costly and time-consuming processes [28,29]. Thus, a platform for validating cloned BE plasmids, including sgRNA components, may save the investment of resources and time. Incidentally, plasmid vectors with the ability of ectopic expression of Cas enzyme and sgRNAs in bacteria will facilitate the quick screening and reliability of cloned plasmids and editing activities of designed gRNAs. Among the analyzed p35S versions (p35S long with or without intron and p35S short), intron-containing p35S labeled as p35S(L)I [17] showed moderate sfGFP expression activity (data not shown). Therefore, two differential-strength promoters, high-strength pGlpT and low-strength p35S(L)I, were selected for nCas9-BE expression in *E. coli*.

We sought to inspect 10 different RNA polymerase III (RNA Pol III)-dependent promoters for conserved −10 (TATAAT) and −35 (TTGACA) elements (Appendix A). The promoters were chosen because of earlier studies reporting efficient sgRNA expression in different species, including bacteria (pJ23119), plants (pAtU6, pOsU3, pOsU6, pMtU6.6, pZmU3, pTaU3, pTaU6), yeast (pSNR52), and human (phU6). The pJ23119 promoter has successfully been used for sgRNA expression in *E. coli* and other bacteria [9,14,30]. We detected partial or fully conserved −10 and −35 elements in all the promoters. Remarkably, the DNA sequence alignment of pJ23119 and pAtU6 showed 54.28% identity with fully conserved −10 and partially conserved −35 elements (Figure 1B). Therefore, in view of developing a heterologous promoter for sgRNA expression, pAtU6 was examined for optimal sgRNA expression in *E. coli*.

#### 2.1.2. Modular Design for Desired Target DNA Cloning and Platform Optimization

We employed a PmCDA1-based Target-AID system that induces C-to-T mutations in the editing window located approximately 1 to 8 bases from the distal end of the PAM (counting positions 21–23 for PAM) in 20 bp protospacer [6]. We adopted two approaches for BE testing. Firstly, a two-plasmid system consisted of a target plasmid and a CRISPR plasmid including Pro-sgRNA and Pro-nCas9(D10A)-PmCDA1-1×UGI (Figure 1C). Secondly, a single-plasmid system was composed of the target, sgRNA, and Pro-nCas9(D10A)-PmCDA1-1×UGI assembled into a single vector (Figure 1D). We analyzed both single- and two-plasmid vector systems, showing similar editing outcomes (Appendix A). Therefore, a single-plasmid system was used for further analysis.

Next, the target regions for the Test sgRNAs were synthesized and cloned into the designed universal target-acceptor module (Figure 2). The intended 23 bp target sequence (20 bp guide + 3 bp PAM) was cloned using BsmBI type IIS sites in such a manner so that codons would be in frame with the sfGFP sequence. The sfGFP sequence facilitates the screening of successfully cloned plasmids depending on the presence or absence of fluorescence. Cloning of the required target DNA region into an sfGFP-based universal target-acceptor using MoClo kit protocol [17] permits the assessment of any intended sgRNA.

Two sgRNAs were tested containing alternate Cs at even (test sgRNA1 target region, Figure 3A) and odd (test sgRNA2 target region, Figure 3B) positions spanning from 1 to 10 in the 20 bp protospacer. After transforming individual constructs in 10-beta *E. coli*, single colonies were cultured for plasmid isolation and target-region sequencing. Base editing frequency (ratio of edited colonies to total screened colonies containing at least one C-to-T in editable window) of nCas9-PmCDA1-1×UGI was found to be 100% in the case of pGlpT and p35S(L)I with variable efficiency in the editing window ranging from 1 to 6 position (Figure 3C,D). Calculated editing efficiency for different Cs within the editable window showed a variable percentage of C-to-T editing, implying that after transforming the plasmid vectors into competent cells, the grown individual clones were indeed mixed populations. Similar outcomes for base editing were previously reported in other bacteria, such as *Corynebacterium glutamicum* [31], *Clostridium beijerinckii* [32], *Staphylococcus aureus* [22], and *Agrobacterium* sp. [14]. For pGlpT, four colonies showed an average of 92% editing at targeted C4 in Test sgRNA1 target region and 79.5% editing at targeted C3 in Test sgRNA2 target region (Figure 3C,D). A variable range of editing efficiencies was observed for each C position in nCas9(D10A)-PmCDA1-1×UGI driven by different promoters, possibly due to differential expression of CRISPR-BE reagents. Moreover, base editing events with a single, clear genotype can be achieved from mixed population by streak plating (Figure 3E). Besides, the nCas9/dCas9-based pGlpT-CBE system together with pJ23119-sgRNA showed no correct clones (Appendix A). The combination of nCas9/dCas9-based pGlpT-CBE and pAtU6-sgRNA permitted sufficient cell survival allowing successful BE analysis. These data indicate that the choice of promoters (driving BE or sgRNA) is a crucial parameter to avoid the lethal effects in BE experiments (Appendix A). Consequently, the optimized method including differential-strength promoters for expression of CRISPR components was named IRI-CCE, an in vivo rapid investigation of CRISPR components in *E. coli*. The IRI-CCE can be expanded by screening more nontoxic promoters with a wide range of strength in the future.

### 2.2. Assessing Base Editing Activities on Chromosomal Targets in E. coli Genome

Next, we targeted *E. coli* genes to evaluate the BE efficacy and potential toxicity mediated by differential-strength promoters in the genomic context of bacteria. Five different sgRNAs were chosen for editing three genes, namely *galK*, *rpoB*, and *rppH* (Figure 4). The 10-beta strain lacks the *galK* locus; therefore, DH5α strain was used for examining the CBE-mediated editing of the *galK* locus. The pGlpT and p35S(L)I-driven nCas9(D10A)-PmCDA1-1×UGI efficiently edited the available Cs in the editing window of all the tested sites with sufficient cell survival for BE investigations.

Because the editing efficiency of BE versions may differ due to distinct genetic backgrounds in different *E. coli* strains, we also examined the cytotoxicity and base conversion efficiency in three more strains, including DH5α, DB3.1, and BL21(DE3) for Test sgRNA1 target region using nCas9(D10A)-PmCDA1-1×UGI. Irrespective of the genetic background of strain, we found sufficient cell survival allowing BE assays (Appendix A) that confirmed comparable C-to-T mutations in tested sgRNAs in all the four bacterial strains (Figure 5), suggesting the broader applicability of the IRI-CCE platform across different *E. coli* strains.

### 2.3. Optimization of Broad-Range CBEs Using IRI-CCE Platform

Base editors with broader editing windows are generally applied for mutagenesis of a user-defined region. The editing windows vary according to BE type, target site, sequence context, and experimental conditions [7]. Several latest BE tools are not yet characterized for *E. coli* use. Mainly, features of evoCDA1, APOBEC3A, ABE8e, and ABE9e with enhanced BE activity in other organisms are not yet analyzed in bacteria. To characterize the editing features in *E. coli*, two CBE types (evoCDA1 and A3A) were tested in the IRI-CCE platform with distinctive features (different range of editing window and sequence context-preference). In original reports, editing windows for evoCDA1 and A3A were positioned from 1 to 14 [18] and 1 to 17 [19] with no sequence context-preference for evoCDA1 and moderate CT preference for A3A. A core-editing region was observed between positions 3 to 12 for evoCDA1 in human cells and positions 3 to 9 for A3A in plants. To examine C-to-T conversion in *E. coli*, in addition to two Test sgRNAs, we designed Test sgRNA3 by mimicking the sequence of TaVRN1-sgRNA1 (Figure 6A) that comprises the Cs in the range of 1 to 17 bp [19].

Constitutive expression of CRISPR components using different promoters showed an expanded editing window for evoCDA1 and A3A, possibly owing to higher nCas9-BE expression. Among the tested CBEs, evoCDA1 showed the C-to-T conversion with the broadened editing window from positions −6 to 14 with pGlpT and from positions −1 to 9 with p35S(L)I (Figure 6B). The expression of nCas9-A3A using pGlpT and p35S(L)I promoter also displayed distinct editing windows ranging from −6 to 17 and 2 to 17, respectively (Figure 6C). For pGlpT-evoCDA1, C-to-T conversion at positions −1 to 9 was the highest (18–100%), whereas for pGlpT-A3A, C-to-T mutation at positions 3 to 10 was the highest (43–100%).

On the other hand, for p35S(L)I-evoCDA1, C-to-T conversion at positions 2 to 7 was the highest (57–100%), whereas for p35S(L)I-A3A, C-to-T mutation at positions 4 to 10 was the highest (38–100%). Notably, G:C at position −5 was mutated to A:T in both the CBE types with moderate efficiency across all the three sgRNAs (up to 43%), implying the C-to-T conversion in the target (non-deaminated) strand. In the case of CBEs, noncanonical BE outcomes at PAM-distal positions in the target region were also reported in human cells [33,34]. The upstream (positions −3 to −1) regions of target sites were not accessible to A3A deamination in the tested combinations of sgRNAs and respective target sites (Figure 6C). In addition, A3A-mediated C-to-T editing at middle positions (12 to 16) in Test sgRNA3 was not observed. The comparison between nCas9(D10A) and the CBE types (PmCDA1, evoCDA1, and A3A) containing *rppH* sgRNA3 targeting *rppH* gene indicated that all CBEs are less toxic with slight variation between pGlpT and p35S(L)I promoter tests (Appendix A) and survived cells allowing BE activity studies. Although direct comparison with earlier reports is not possible due to earlier mentioned parameters, these findings demonstrate the robust applicability of the IRI-CCE platform to investigate features of broad-range CBE tools.

### 2.4. ABE8e and ABE9e Activities for A-to-G Conversion in E. coli

Recently, evolved ABE variants, TadA-8e (ABE8e) and TadA-9e (ABE9e), have been characterized and have shown improved A-to-G base conversion in vitro, in humans, and in plants, respectively [20,21], but have not yet been systematically evaluated in bacteria. The sgRNAs with the multiple As in a target region exhibited different A-to-G editing efficiencies for each position in *E. coli* and *S. aureus* [22]. Moreover, the editing window for ABE was also reported to vary with the target site, host species, genotype, and experimental conditions [7,35]. We sought to investigate the ABE8e and ABE9e activity using two synthetic sgRNAs (Figure 7). The editing window for ABE8e and ABE9e variants was reported as positions 4 to 8 in human cells [20] and 1 to 11 in plants [21], respectively. Editing levels of ABE8e and ABE9e showed significant increases in the case of pGlpT compared to that of p35S(L)I (Figure 7A–C).

We observed the editing window of ABE8e from positions 3 to 8 and 5 to 7 for pGlpT and p35S(L)I, respectively. The editing window of ABE9e for pGlpT and p35S(L)I ranged from positions 3 to 8 and 3 to 7, respectively. A-to-G conversion at positions 5 to 7 was the highest in the case of both the variants (ABE8e: 74–100%; ABE9e: 99–100%). For p35S(L)I-ABE8e, A-to-G mutation at positions 5 to 7 was the highest but in the lower range (21–33%), and p35S(L)I-ABE9e showed a moderate editing activity (55–66%). These data suggest that the editing activity of ABE9e was more enriched than that of ABE8e in *E. coli*, and editing activity increases in a promoter-strength-dependent manner. In the case of ABEs, editing activity outside the gRNA window was not observed. Overall, all CBE and ABE data indicate that the differential-strength promoters may contribute to the variable editing outcomes within the canonical editing window.

### 2.5. IRI-CCE Platform Allows Screening of Functional sgRNAs

Prescreening of sgRNAs before commencing GE studies is crucial in higher eukaryotes to avoid loss of time for sgRNA screening, lower editing activities, and waste of resources. Given that the efficiencies predicted by in silico tools and the actual GE activities may differ significantly, in vitro or preferentially in vivo validation of chosen sgRNAs is vital before the actual tests [36]. Generally, protoplast and transient agroinfiltration assays are used for preliminary screening in plants, with some limitations elaborated in the discussion section. In that scenario, the IRI-CCE platform may complement these existing platforms, allowing the simultaneous evaluation of designed bioparts and functionality of the selected sgRNAs that enable the potential avoidance of incorrect plasmids and inefficient gRNAs. In some instances, sgRNAs predicted as inactive by bioinformatic tools or design criteria were reported to work efficiently under in vivo conditions [37]. On the other hand, even if in silico parameters are not promising for a particular set of sgRNAs, one needs to choose those sgRNAs for editing a specific region of targeted locus due to PAM constraints.

We analyzed sets of active and inactive sgRNAs reported in the previous studies (Figure 8A). Based on the A or C availability in the canonical editing window, we opted for a particular BE type to edit the target sites of individual sgRNAs. The target regions, including the PAM, were cloned using two BsmBI type IIS sites in the universal target-acceptor modules. Firstly, the set of three active sgRNAs (AtGL1-sgRNA1, [38] SlMlo1-sgRNA2, and SlPelo-sgRNA1 [39]) targeting three genomic targets showed higher base conversion efficiencies (Figure 8B–D) with differential-strength promoters as observed under in vivo conditions.

Secondly, we analyzed ABE8e activities for two inactive sgRNAs (Inactive sgRNA1 and Inactive sgRNA2) that displayed poorer performance [40]. The lower editing activity was attributed to internal sgRNA interactions that allow binding with Cas9, but it prevents target DNA recognition by the Cas9-sgRNA complex (Appendix A). Besides, inactive sgRNA competes with active sgRNAs for binding to the Cas9 protein, thereby reducing the editing activities. We investigated the base conversion activity at target sites of both the sgRNAs and discovered that both the sgRNAs led to no BE frequencies in the target DNA region with low-strength p35S(L)I (Figure 8E,F), consistent with the in vivo data for CRISPR/Cas editing [40]. Moreover, even a high-strength promoter (pGlpT)-driven ABE8e expression led to relatively lower BE activity (36% and 23.5% for Inactive sgRNA1 and -2, respectively). Overall, these data indicate that the IRI-CCE platform is helpful to distinguish the functional sgRNAs, although with limited information (elaborated in the discussion section). Taken together, the IRI-CCE platform could still be an asset to provide first-hand knowledge of in vivo sgRNA activity one needs to further validate or use in higher eukaryotes. Especially, IRI-CCE may be more sensitive than in planta methods and could be helpful in screening gRNAs for plant species without readily available protoplast or agroinfiltration systems.

### 2.6. Expanding the Targeting Scope of BEs by Altering Editable Window Size Using Different Lengths of Protospacers

Different lengths of protospacers have been reported to alter the editing window of CBE versions in the case of plasmid and chromosomal targets [9,41,42]. To further characterize the effect of protospacer length together with promoter strength on editing window shift, protospacers 18, 20, 21, 22, 23, 25, and 30 nt in length were tested for CBE (Target-AID, Figure 9A) and ABE (ABE9e, Figure 9B). A narrow editing window was observed for truncated protospacer (18-nt) compared to conventional 20-nt protospacer. The editable window was shifted towards the 3′-end of the target sequence, i.e., towards the PAM site (counting PAM as 21 to 23). In contrast, sgRNAs with extended length consistently showed window shift and expansion of the editable window towards the 5′ end of the target sequences in the case of both the promoter types (Figure 9).

As shown in Figure 6, broad-range CBEs, particularly evoCDA1, exhibited higher C-to-T conversion with a wider editing window in 20-nt protospacer (pGlpT: positions −6 to 14, p35S(L)I: positions −1 to 9). Likewise, higher editing was observed in all the tested protospacers of different lengths for pGlpT and p35S(L)I promoters for evoCDA1 (Figure 10). Particularly at −1 and −3 positions for p35S(L)I, base editing was about 4-fold and 30-fold higher than with a 20-nt protospacer. With a longer length of protospacers (21, 22, 23, and 25 nt), C at position −6 was edited (up to 19%), which was unchanged in 20-nt protospacer for p35S(L)I. Interestingly, 30-nt protospacer was found to be less active in all three BE versions. A similar trend was reported for protospacers with lengths of more than 25 nt [41]. Overall, BE tools showed narrow or wider editable windows in the case of differential-strength promoters together with extended or truncated protospacers.

## 3. Discussion

Several features of *E. coli* make it a model microorganism in molecular biology. These include a short life cycle, simple genetic manipulations, low cost, and in-depth knowledge of the genome [43]. In the current study, we developed an *E. coli*-based IRI-CCE platform to investigate important aspects, including optimization of BE tools, validation of BE bioparts for further use in modular cloning approach, and sgRNA functionality with several advantages. IRI-CCE enables the rapid verification of designed BE bioparts. Modular bioparts are exchangeable for cloning and investigating features of different CRISPR-based BE tools in bacteria and plants, including deaminase variants, Cas9 forms, heterologous promoters (p35S and pAtU6), and accessory components such as UGI. Moreover, the approach reported here could be readily scaled to many constructs and different BE types and evolved versions. In addition, multiple gRNA and DNA targets can be evaluated by using modular cloning assembly in the future. In addition, IRI-CCE could also be useful for testing novel BE architectures.

The attempt to analyze the BE activity of bacterial (pJ23119) promoter-driven sgRNA expression was unsuccessful due to the toxicity caused by BE components. Likewise, the pEc1-mediated CBE test also showed poor transformation efficiency with a mix of incorrect clones. The higher expression of BE components adversely affects cell survival in bacteria [9,14]. When nCas9(D10A)-PmCDA1-1×UGI was expressed together with pJ23119-sgRNA, no correct clones were obtained in *E. coli*, possibly due to toxic effects of BE components [9]. However, nCas9(H840A)-PmCDA1 did not lower the transformation efficiency, suggesting that the SSBs induced by nCas9(H840A) may be insufficient to cause cell toxicity. Instead, more likely, the transient DSBs generated by two nicks on different strands by nCas9(D10A)-PmCDA (one on the non-deaminated strand by nCas9(D10A) and the other on the deaminated strand by AP endonuclease during the repair process) may lead to cell toxicity. A high rate of SSBs and transient DSBs induced by nCas9(D10A)-PmCDA1 during pJ23119-mediated sgRNA production seems challenging to restore by the bacterial system, ultimately resulting in losing the plasmid. The cloning of dCas9-PmCDA1-1×UGI with pJ23119-sgRNA showed the wrong clones in *E. coli* [9], suggesting cells cannot retain the functional dCas9-PmCDA1-1×UGI-expressing plasmids. Although dCas9 does not introduce SSBs or transient DSBs, earlier dCas9-PmCDA1-1×UGI data indicated that the surplus amount of UGI compromises the genome integrity in CBE types [9].

Previous studies reported that the use of protein degradation (LVA) tag fusion with dCas9-Target-AID reduced the toxicity to *E. coli* cells [9]. Producing the optimal amount of BE and sgRNA might reduce the toxicity to a tolerable level for a bacterial cell, as observed in nCas9-BE together with pAtU6-driven sgRNA expression. As described in our study, the combination of pGlpT-nCas9-BE or p35S(L)I-nCas9-BE with pAtU6-sgRNA expression units proved to be the most appropriate to edit the target site in the same plasmid vector or *E. coli* genome without causing cytotoxicity. Therefore, fine-tuning the expression of BEs and sgRNAs by using the optimal combination of promoters is a crucial factor in minimizing the toxic effects in the IRI-CCE platform. A recent study [44] indicated that the sufficiently perturbed dsDNA in bacteria—during replication, transcription, or other rearrangements—provides enough opportunity for Cas9 binding to a target site in the form of dynamically stretched single-strand DNA (ssDNA) or un- and underwound double-strand DNA. Therefore, it is conceivable that the fewer nCas9-BE enzyme molecules, mainly evolved versions with a faster catalytic rate [18,45], produced by p35S-based promoters are enough to show relatively higher deamination activity.

Deaminase interaction with Cas protein and target DNA region alters the location and size of the editing window [46]. The indispensable role of the endogenous DNA repair mechanism is a decisive factor that may produce variable BE results in different organisms [47]. Nevertheless, we observed significant differences between the targeting competencies of three CBEs owing to the differential strength of promoters used for Cas-BE expression (Appendix A). The evoCDA1 is an evolved variant of PmCDA1 [18] that showed higher catalytic activity and expanded editing window than its wild-type counterpart. We observed higher C-to-T editing in the expanded window up to position −6 that was not detected in the previous work for evoCDA1 or A3A. It has been reported that different lengths of gRNA enable the adjustment of the editing window in BE tools [9,41,42]. Likewise, tested BE tools showed narrow or wider editing windows in the case of differential-strength promoters and extended or truncated gRNAs (Figure 9 and Figure 10). Therefore, we propose using a combination of variable lengths of sgRNAs, differential expression of BE components, and engineered Cas9 variants with relaxed PAM sites as a new approach to adjust the editing window and targeting scope of BE tools.

Although we consistently observed differences in the editing window size for all the tested BE versions under differential-strength promoters (Appendix A), the exact mechanism is unclear. We note that a recent report by Hao and coworkers [41] has analyzed the effects of variable expression levels of nCas9-PmCDA1 on conversion efficiency and editing window in *B. subtilis*. Higher base editing (C-to-T) efficiency and wider editing window were exhibited with an increased concentration of inducer in the case of xylose promoter (P_xyl_). Notably, the most preferred position is the same for specific BE versions irrespective of the promoter type (PmCDA1, position 4; evoCDA1, positions 2 to 7; APOBEC3A, positions 6 to 8; ABE8e, positions 5 to 7; ABE9e, positions 5 to 7). Nucleotide conversion efficiency on both sides of the most preferred position showed a gradual decrease in all tested promoter–BE combinations, consistent with previous reports about enzymatic activities of deaminases [45,48,49]. Indeed, the interplay between various factors related to enzyme–substrate concentrations (nCas9-BE, ssDNA) inside the cell may contribute to the final BE outcome. Some key aspects involve the rate of DNA replication, target DNA packaging and accessibility, amount of nCas9-BE, the kinetics of deamination, and DNA repair of deaminated nucleotides [48,50]. Further studies are needed to elucidate the role of these factors in BE outcomes. We believe that further comprehensive analysis of the promoter-strength-dependent variability in the editing window will help to identify the factors responsible for wider or narrow editing windows.

Some studies reported C-to-G or C-to-A as CBE editing byproducts [5,8]; we did not observe this phenomenon in the case of basic (PmCDA1, APOBEC3A) or improved CBE (evoCDA1) versions, which might be too low to detect with Sanger sequencing if present. The evoCDA1 and APOBEC3A exhibited different editing windows, possibly due to differences in editing competencies of deaminases and target accessibility during R-loop formations. Some Cs in the editing window were not accessible to APOBEC3A deamination. These observations highlight the importance of considering BE type and promoter strength that best suit the expected outcome. Moreover, temperature affects the enzymatic activities of CRISPR components, thereby influencing the GE outcome [2]. Mainly, the performance of Cas enzymes is best at 37 °C. So, besides the promoter activity and evolved variants, better editing efficacy in IRI-CCE may result from the favorable temperatures for enzymatic activities.

The IRI-CCE system in its present form has some limitations related to validating plant-related factors while optimizing BE modules or gRNA screening. Some plant-related factors cannot be assessed in this bacteria-based IRI-CCE system for reasons that include the effect of genome organization on target accessibility, the feature to evaluate nuclear localization signal (NLS) function (although NLSs are included in the expression modules), different DNA repair machinery in bacteria, and difference in growth temperature of bacteria and plants. However, IRI-CCE offers an efficient platform for bacterial BE studies, evidenced by its successful use for optimizing different nCas9-based BE versions (PmCDA1, evoCDA1, APOBEC3A, ABE8e, and ABE9e), which is also the first report in *E. coli* to the best of our knowledge. In addition, IRI-CCE allows verification of the sgRNA-related factors such as nCas9-BE-sgRNA complex formation and target DNA-binding irrespective of the host system. Some existing platforms can be helpful for prescreening of gRNAs and preliminary check of CRISPR-BE outcomes. For instance, the combination of protoplast culture and preassembled ribonucleoprotein (RNP) complexes is one of the handy methods for assessing the editing activities of sgRNAs in plants [51].

Protoplast systems are relatively easy to use as testbeds for GE tests. A growing number of protocols are being established for protoplast isolation and RNP-mediated delivery of CRISPR reagents in many plant species, including model and nonmodel plants [52,53,54,55,56]. Nevertheless, the prerequisite of protoplast-based assays is the establishment of optimized protoplast isolation procedures for the crop of interest and purified Cas-sgRNA RNP complexes, which further adds to the cost. Transient agroinfiltration assay proved to be a comparatively quick in vivo method for analyzing sgRNA efficacy, but the lower activity and inconsistency depending on target plant species are the significant obstacles [57]. The IRI-CCE platform may serve as a fast way to validate some key aspects. For instance, the sgRNA-related factors such as nCas9-BE-sgRNA complex formation or stability and target DNA-binding would be identified in the IRI-CCE platform. We propose that the users choose the moderate p35(L)I promoter in sgRNA prescreening experiments to avoid overestimating sgRNA activity while using the pGlpT promoter. Moreover, the use of a specific BE type depending on the nucleotide composition of the target region is recommended. However, more efforts are needed to understand uncovered aspects. For example, the editing efficiency of each nCas9-BE-sgRNA complex differs significantly depending on the target locus [39]. In higher eukaryotes, editing differences arise from the sequence bias in DNA repair machinery and target accessibility [58]. Moreover, the bigger genome size of eukaryotes influences the dynamics of target search. Therefore, IRI-CCE is missing those features common to eukaryotic cells and cannot be verified in the current platform. While our data show the promising way to confirm the biopart modules for CRISPR-based BE work, there are multiple opportunities to expand the IRI-CCE utility further in other bacteria. Moreover, the design and adoption of new dual BEs by combining different variants of ABE and CBE could be interesting for simultaneous conversion of A to G and C to T, respectively.

## 4. Materials and Methods

### 4.1. E. coli Strains

The following four *E. coli* strains were used: 10-beta, DH5α, DB3.1, and BL21(DE3). The information about strain genotype is provided in Appendix A. The primers, plasmids, and biopart sequences used in this study are listed in Appendix A, respectively. The *E. coli* strains were cultured in Luria-Bertani (LB) broth with appropriate antibiotics.

### 4.2. Plasmid Construction and Cloning

The required bioparts were amplified by conventional polymerase chain reaction (PCR) using a Phusion high-fidelity DNA polymerase (Thermo Fisher Scientific, Waltham, MA, USA). The cloning of different plasmid vectors was designed and performed by following the principle of MoClo [59] and Golden Gate assembly [17] protocols using BpiI/BsaI Type IIS enzyme digestion–ligation. The DNA recognition sites of restriction enzymes BsaI and BpiI were removed from internal sequences of bioparts during PCR amplification to make them suitable for Golden Gate cloning (a procedure also termed domestication). The DNA oligonucleotide pairs for the pEc1 (SJM901 + RBSTL2) promoter and TerL3S2P21 terminator were annealed and ligated into BpiI-digested acceptor plasmids. Other promoters including pGlpT and p35S(L)I were used from the pYTK001 (Addgene #65108) [25] and pICH51266 (Addgene #50267) [17], respectively. The promoter DNA sequences and their evaluation are provided in Appendix A. Similarly, DNA sequences of Ter35S Terminator were adopted from the pICH41414 (Addgene #50337) [17].

Three cytidine deaminases, namely PmCDA1 (Target-AID) [6], evoCDA1 [18], and APOBEC3A [19], were PCR amplified from PmCDA1-1×UGI (Addgene #79620), evoCDA1 pBT277 (Addgene #122608), and A3A-PBE-ΔUGI (Addgene #119770), respectively. For all three deaminases, the same linker regions were used as reported in the previous studies. The ABE8e was synthesized from Bioneer Co. (Daejon, Korea), and the ABE9e variant (containing additional V82S/Q154R mutations) was cloned using ABE8e module as a template by site-directed mutagenesis PCR. PmCDA1-1×UGI was fused to the C terminus of nCas9. The evoCDA1, A3A, and ABE8e were fused to the N terminus of nCas9 with the XTEN linker. The 2×UGI module (template: Addgene #122608) was fused to the C terminus of evoCDA1-nCas9 and A3A-nCas9. The sgRNA expression was driven by bacterial pJ23119 (synthetically cloned) or plant AtU6 promoter (pICSL01009, Addgene #46968). The nCas9(D10A) was generated by the PCR method using previously optimized Cas9 as a template (Level 1 hCas9 module, Addgene #49771). Desired sgRNA sequence was PCR amplified using plasmid pICH86966::AtU6p::sgRNA_PDS (Addgene #46966) as a template and cloned together with either pJ23119 or pAtU6 for sgRNA expression. All the Addgene constructs ordered from https://www.addgene.org/, accessed on 9 December 2021.

For cloning of the target region of the desired sgRNA sequence, a pair of oligonucleotide DNAs that contained the target sequence with PAM was annealed and ligated into BsmBI-digested universal target-acceptor plasmid L1 or L2 having optimized sfGFP [25] at downstream side (Figure 1 and Figure 2).

### 4.3. Bacterial Transformation, Plasmid Isolation, and Sanger Sequencing

Ligation products of all the steps during the cloning were transformed into competent cells of *E. coli* 10-beta strain by a heat-shock method. The bacterial culture was spread on the LB media containing desired antibiotic and incubated at 37 °C for 18–24 h. The cloned plasmids and BE activities were confirmed by Sanger sequencing at Solgent Ltd. (Daejeon, Korea) or Cosmogentech Ltd. (Seoul, Korea). For mutagenesis assay, the different *E. coli* strains were transformed with the appropriate plasmids using the heat-shock method and were precultured for 1 h with 1 mL of LB medium. After incubation for 1 h at 37 °C, the fraction of cell cultures were spread on LB agar (1.5%) plates containing selection antibiotics with needed concentrations. The next day, individual colonies from the plate were inoculated in 3 mL LB broth with appropriate antibiotics and cultured at 37 °C. The culture time was varied according to the experimental parameters and mentioned in appropriate sections. The plasmid isolation was performed using Plasmid Mini-Prep Kit from BioFact Co. Ltd. (Daejeon, Korea) for Sanger sequencing analysis.

### 4.4. Promoter Activity Analysis

RNA was extracted using the following method: *E. coli*-carrying plasmids (Figure 1B) were grown in LB medium. Cells were harvested during the exponential growth phase (12 h with OD_600_ value 0.5). Total RNA was extracted using RNeasy Protect Bacteria Mini Kit from Qiagen. For all samples, 600 ng of total RNA was used for complementary DNA (cDNA) synthesis using a QuantiTect Reverse Transcription Kit from Qiagen following the manufacturer’s instructions. To estimate the relative *sfGFP* transcript, the quantitative real-time PCR (qRT-PCR) reactions were carried out using the KAPA SYBR FAST qPCR kit from Kapa Biosystems (Wilmington, MA, USA) with *sfGFP*-specific primer sets (Appendix A). Cycling of PCR consisted of pre-denaturation at 95 °C for 5 min followed by 40 cycles of a denaturation step at 95 °C for 10 min, an annealing step at 60 °C for 15 s, and final extension step at 72 °C for 20 s using the CFX384 Real-Time System from Bio-Rad (Hercules, CA, USA). The qRT-PCR reactions were performed with independent biological replicates. Relative *sfGFP* transcript values normalized against internal control 16S ribosomal RNA (*rrsA*) gene. Data analyses were performed by the 2^−ΔΔCt^ method [60].

### 4.5. Evaluation of Editing Activities

The single colonies were cultured, and the plasmid vectors containing synthetic targets were purified using Plasmid Mini-Prep Kit from BioFact Co. Ltd. (Daejeon, Korea) for further sequencing analysis. For sequencing analysis of genomic loci, the target fragments were PCR amplified using target-specific primers from the randomly picked colonies and then analyzed by Sanger sequencing. Sanger sequencing data analysis was performed using SnapGene software version 3.2.1 (GSL Biotech; available at snapgene.com, accessed on 9 December 2021). The editing efficiency was determined by the ratio of nonedited to edited colonies from the randomly picked cells. The base conversion rate was estimated using the online tool EditR [61] from C to T and A to G for CBE and ABE, respectively. The data were statistically analyzed and plotted in GraphPad Prism software version 9.0.0 for windows from GraphPad Software, San Diego, CA, USA (www.graphpad.com, accessed on 9 December 2021).

## 5. Conclusions

In summary, nCas9-BE components expressed by promoters of different strengths led to the establishment of the nontoxic IRI-CCE platform for two major applications: investigating cloned CRISPR-BE components and understanding sgRNA functionality. IRI-CCE analyses showed the variable length of the editing window for specific BE types. Each independent BE type has its specific editing pattern for a given target site and promoter type. Because of its easy cloning and heterologous applicability, the IRI-CCE platform may be suited for rapid evaluation of cloned bioparts in modular cloning of BE studies and applicable to a broader range of *E. coli* strains being used for cloning. This platform facilitates the verification of sgRNA-related factors such as cloning, sgRNA production, Cas9-sgRNA complex formation, and DNA recognition. Through the applications described here and through further improvements, IRI-CCE can be widely applicable for the characterization of BE components and rapid assessment of newer CRISPR-based BE technologies.

## Figures and Tables

**Figure 1 ijms-23-01145-f001:**
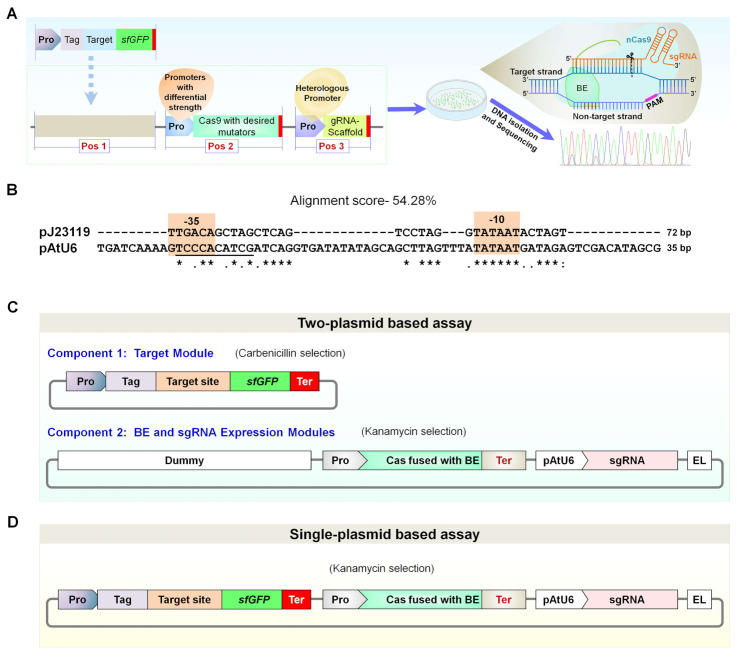
Schemes used for optimization of in vivo rapid investigation of CRISPR-BE components in *Escherichia coli* (IRI-CCE) platform. (**A**) Schematic representation of IRI-CCE platform. Promoters of differential strength may enable the production of CRISPR-BE components (Cas9-fusions and sgRNAs) without toxic effects in *E. coli*, allowing the testing of new BE tools, reliability of cloned plasmids, and editing activities of designed sgRNAs. (**B**) Sequence alignment of bacterial pJ23119 and *Arabidopsis* U6 (AtU6) promoter. The features of AtU6 promoter include two conserved elements: the upstream sequence element (USE; consensus sequence RTCCCACATCG) and a TATA-like box (consensus sequence TTTATATA) which are highlighted using underline and black box, respectively. “*” denote identical nucleotides, “:” denote conserved nucleotides, and “.” symbol denote semi-conserved nucleotides. (**C**) Two approaches analyzed for the establishment of base-editing-based IRI-CCE platform. Two-plasmid-based assay comprises two independent plasmids. The first plasmid consisted of the target-region-expression plasmid with C-terminal superfolder green fluorescent protein (sfGFP). The second plasmid consists of an assemblage of promoter-sgRNA and promoter-nCas9-PmCDA1-linker-1×UGI-terminator (nCas9-PmCDA1) in a plasmid vector. (**D**) A single-plasmid assay system composed of all three parts assembled into a single plasmid, including target region, promoter-sgRNA, and nCas9-PmCDA1.

**Figure 2 ijms-23-01145-f002:**
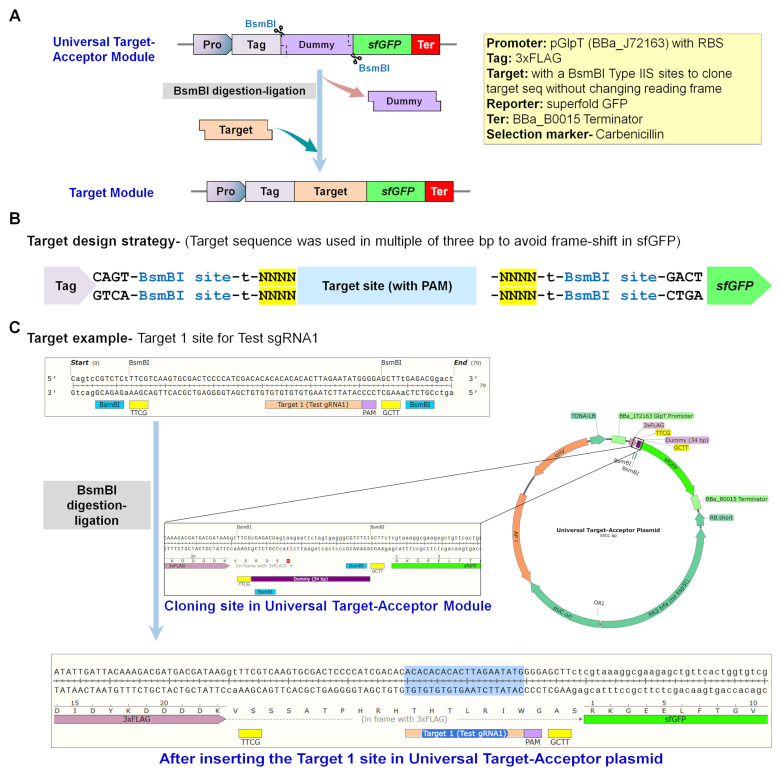
Scheme followed for cloning the target regions as DNA templates for sgRNA binding and subsequently CRISPR-BE editing. (**A**) Schematic representation of cloning region in the universal target-acceptor module. Different components used for assembly are summarized in the yellow box. Cloning of the desired target region into a universal target-acceptor through type IIS (BsmBI) enzyme digestion–ligation permits the assessment of any intended sgRNA. (**B**) The intended 23 bp target sequence (20 bp guide + 3 bp PAM) was cloned using BsmBI type IIS sites in such a manner so that codons would be in frame with the sfGFP sequence. The sfGFP sequence facilitates the screening of successfully cloned plasmids depending on the presence or absence of fluorescence. (**C**) Example of target cloning for Test sgRNA1 is shown.

**Figure 3 ijms-23-01145-f003:**
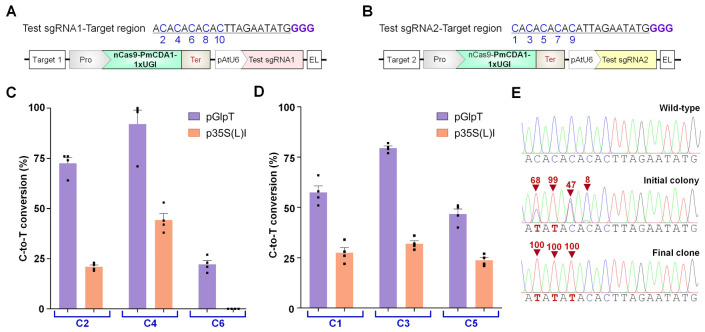
Evaluation of C-to-T conversion by PmCDA1-mediated cytosine base editor (CBE) driven by different promoters in *E. coli* cells. (**A**,**B**) Two independent sgRNAs were tested containing alternate Cs at even (Test sgRNA1, (**A**)) and odd (Test gRNA2, (**B**)) positions spanning from 1 to 10 in the 20 bp protospacer length. The numbers are assigned by counting the distal base as 1 from the protospacer adjacent motif (PAM), i.e., GGG. Schematic representation of the plasmid vectors showing a synthesized target region for sgRNA and two transcriptional units (TUs) composed of nCas9 (D10A) fused with PmCDA1-1×UGI and AtU6 promoter-sgRNA unit. (**C**,**D**) PmCDA1-based C-to-T editing activities. The base conversion rate was estimated using the online tool EditR. Graph values show the mean percentage on the *y*-axis and the tested protospacer positions on the *x*-axis. The graph bar shows the mean of percentage values, and error bars indicate the standard error of the mean (mean ± s.e.m.) of four independent biological replicates. Dots indicate the individual biological replicates. (**E**) Sanger sequencing data of Test gRNA1 targeting region from the colony obtained directly after transformation (initial colony). Sanger sequencing data obtained after streak plating of initial clone culture (final clone). The substituted base is marked with a red inverted triangle.

**Figure 4 ijms-23-01145-f004:**
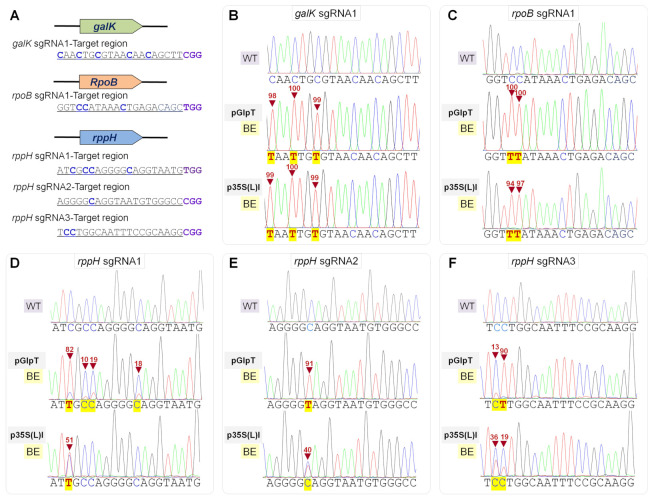
Differential-strength promoter-driven C-to-T editing activities by PmCDA1 in *E. coli* genome. (**A**) Designed sgRNAs and PAM site (violet) for targeting three genes in the *E. coli* genome. Out of five sgRNAs, 1, 1, and 3 were used for targeting *galK*, *rpoB,* and *rppH*, respectively. All the cytosine nucleotides located in the protospacer are highlighted in blue (bold). (**B**–**F**) Sequence alignment of the PmCDA1-edited mutants. The pGlpT and p35S(L)I-driven Target-AID efficiently edited the available Cs in the editing window of all the tested sites. Notably, in the p35S(L)I-driven Target-AID system, higher C-to-T activity was observed in *galK* and *rpoB*, which might be attributable to the limited number of available Cs in the editing window compared to multicopy plasmid-encoded targets. The substituted base is marked with a red inverted triangle.

**Figure 5 ijms-23-01145-f005:**
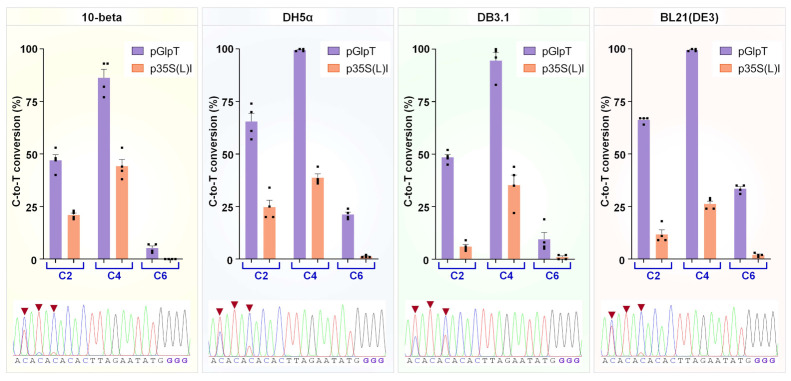
Evaluation of C-to-T conversion by PmCDA1-mediated cytosine base editor driven by pGlpT and p35S(L)I in *E. coli* strains: 10-beta, DH5α, DB3.1, and BL21(DE3). C-to-T editing activities at the synthetic target of Test gRNA1 were examined. Graph values show the mean percentage on the *y*-axis and the tested protospacer positions on the *x*-axis. The graph bar shows the mean of percentage values, and error bars indicate the standard error of the mean (mean ± s.e.m.) of four independent biological replicates. Dots indicate the individual biological replicates. Representative example of Sanger sequencing data of Test gRNA1 targeting region including GGG as a protospacer adjacent motif (PAM) is shown on the bottom side of each panel. The substituted base is marked with a red inverted triangle.

**Figure 6 ijms-23-01145-f006:**
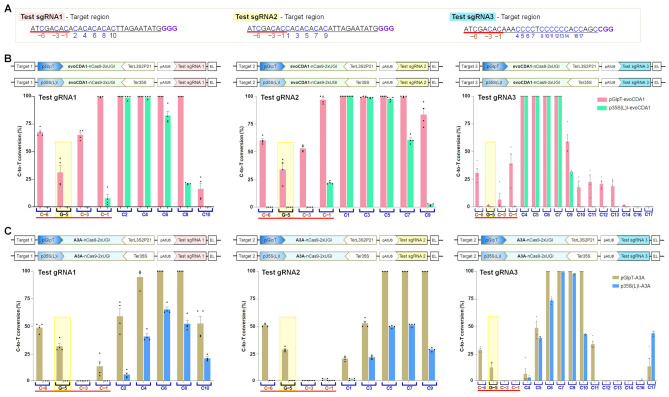
The evoCDA1 and APOBEC3A expressed under the control of different promoters show expanded editing windows with efficient C-to-T editing. (**A**) Nucleotide sequences of three sgRNAs (underlined in black) along with additional *N*-terminal regions (underlined in red). The numbers are assigned by counting the distal base as 1 from the protospacer adjacent motif (PAM), i.e., GGG. (**B**,**C**) evoCDA1 (**B**) and APOBEC3A (A3A) (**C**) driven C-to-T editing activities for three substrates targeted in three independent experiments. The designed BE constructs are represented on the upper side and obtained results are depicted in bar graphs. G-5 position (yellow shade) data showed G·C to A·T conversion, suggesting editing happened in the target strand. Graph values show the mean percentage on the *y*-axis and the tested protospacer positions on the *x*-axis. The graph bar shows the mean of percentage values, and error bars indicate the standard error of the mean (mean ± s.e.m.) of four independent biological replicates. Dots indicate the individual biological replicates.

**Figure 7 ijms-23-01145-f007:**
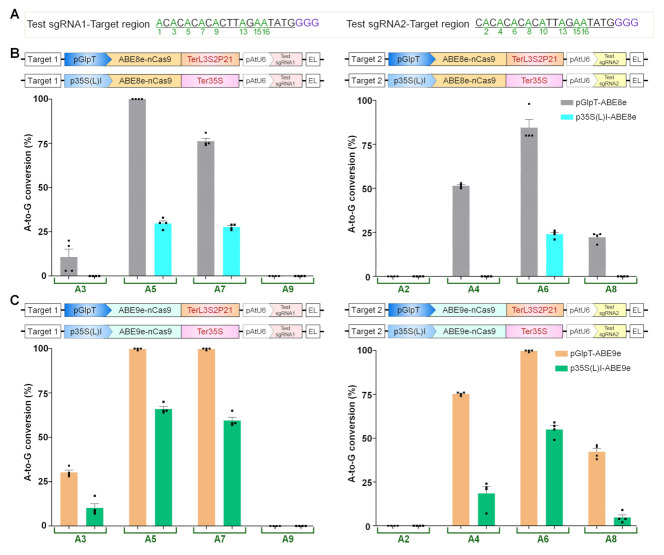
Adenine base editor variants (ABE8e and ABE9e) efficiently convert A to G in *E. coli* cells. (**A**) Nucleotide sequences of Test sgRNA1 and -2 (underlined in black) showing available adenine (**A**) residues (green). (**B**,**C**) The A-to-G conversion efficiencies of ABE8e (**B**) and ABE9e (**C**) driven by pGlpT and p35S(L)I promoters in the target region for Test sgRNA1 and Test sgRNA2. The graph bar shows the mean of percentage values, and error bars indicate the standard error of the mean (mean ± s.e.m.) of four independent biological replicates. Dots indicate the individual biological replicates.

**Figure 8 ijms-23-01145-f008:**
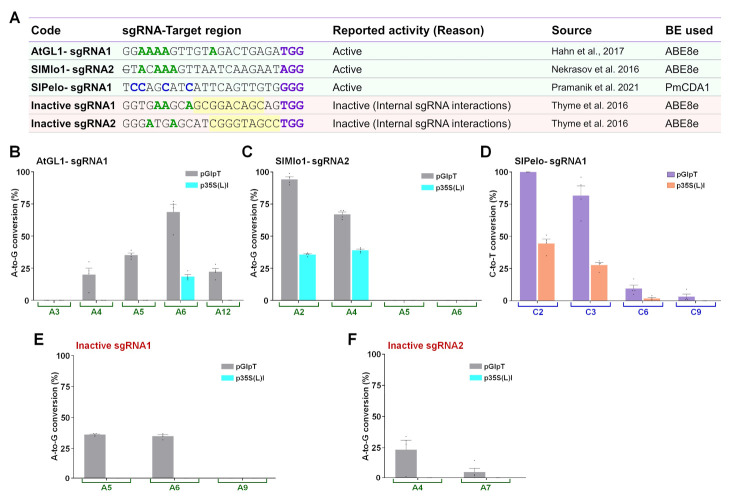
Screening of designed sgRNAs in the IRI-CCE platform provides first-hand knowledge about the sgRNA functionality. (**A**) Set of sgRNAs used for validating the potential use of IRI-CCE in sgRNA screening. The base editor (BE) type was chosen depending on the availability of A or C in the editing window. Frequencies of base conversions by utilized BE types in target regions of AtGL1-sgRNA1 (**B**), SlMlo1-sgRNA2 (**C**), and SlPelo-sgRNA1 (**D**), Inactive sgRNA1 (**E**), and Inactive sgRNA2 (**F**). The graph bar shows the mean of percentage values, and error bars indicate the standard error of the mean (mean ± s.e.m.) of four independent biological replicates. Dots indicate the individual biological replicates.

**Figure 9 ijms-23-01145-f009:**
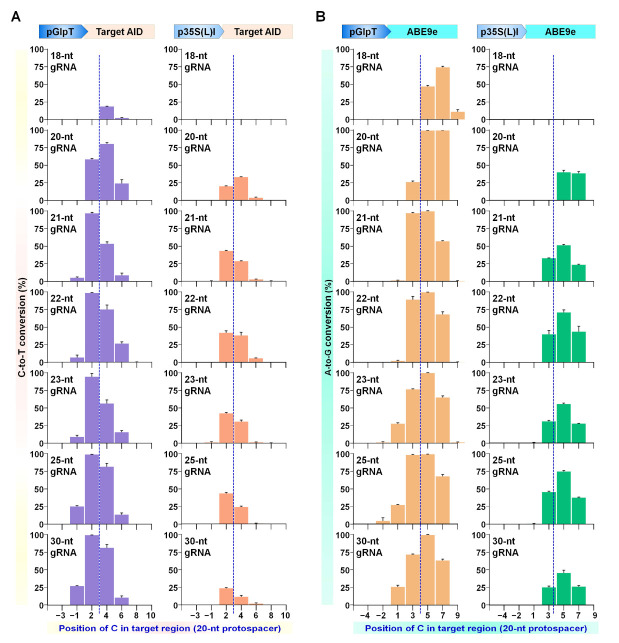
Effect of gRNA length on editing window length in Target-AID and ABE9e. Base editor (BE) efficiencies were tested in the target region by different lengths for Test sgRNA1 (18, 20, 21, 22, 23, 25, 30 nucleotides). Base conversion from C to T with cytosine BE variant Target-AID (**A**) and from A to G with adenine BE variant ABE9e (**B**) driven by pGlpT and p35S(L)I promoters plotted in graphs. The graph bar shows the mean of percentage values of four independent biological replicates.

**Figure 10 ijms-23-01145-f010:**
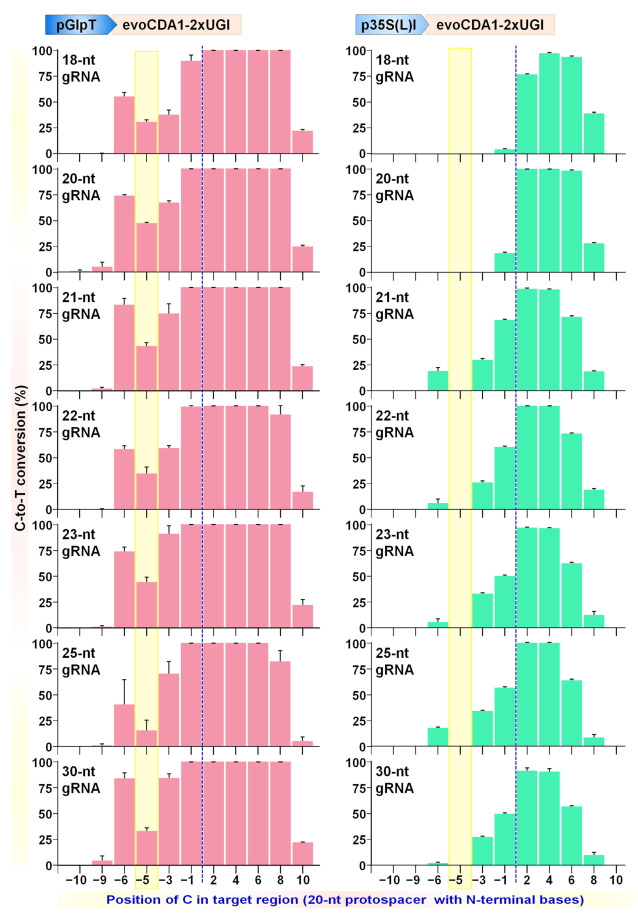
The gRNA lengths alter editing window size in broad-range cytosine base editor evoCDA1. Cytosine base editor variant evoCDA1 expressed using pGlpT and p35S(L)I promoters for C-to-T editing in the target region by different lengths for Test gRNA1 (18, 20, 21, 22, 23, 25, 30 nucleotides). The G-5 position (yellow shade) data showed G·C to A·T conversion, suggesting C-to-T editing happened in the target strand. The graph bar shows the mean of percentage values of four independent biological replicates.

**Table 1 ijms-23-01145-t001:** Toxicity of base editor (BE) components reported in bacterial base editing studies. Editing window numbers counted from the distal of PAM in 20 bp spacers. Promoter-BE reagent combinations exhibiting toxic effects on cell survival are highlighted in gray.

Species	gRNA Promoter	BE Type	Cas9	BE Promoter	Construct Scheme	Editing Window	Cell Toxicity	References
*Escherichia coli*	J23119 (Const.)	PmCDA1	nCas9	Lambda (Induc.)	Lambda-nCas9-3xFLAG-SH3Link-PmCDA1-1×UGI	-	Toxic	[9]
	J23119 (Const.)	PmCDA1	dCas9	Lambda (Induc.)	Lambda-dCas9-SH3Link-3xFLAG-PmCDA1-1×UGI	-	Toxic	
	J23119 (Const.)	PmCDA1	dCas9	Lambda (Induc.)	Lambda-dCas9-3xFLAG-PmCDA1-1×UGI-LVA.Tag	1 to 5 #	Nontoxic	
*Pseudomonas putida* KT2440	J23119 (Const.)	rAPOBEC1	nCas9	Pbs (Const.)	Pbs-rAPOBEC1-nCas9	-	Toxic	[10]
	J23119 (Const.)	rAPOBEC1	enCas9 *	Pbs (Const.)	Pbs-rAPOBEC1-enCas9	-	Toxic	
	J23119 (Const.)	rAPOBEC1	nCas9	ParaBAD (Induc.)	ParaBAD-rAPOBEC1-nCas9	-	Toxic	
	J23119 (Const.)	rAPOBEC1	enCas9	ParaBAD (Induc.)	ParaBAD-rAPOBEC1-enCas9-1×UGI	3 to 8	Nontoxic	
	J23119 (Const.)	rAPOBEC1	enCas9-NG	ParaBAD (Induc.)	ParaBAD-rAPOBEC1-enCas9-NG-1×UGI	3 to 8	Nontoxic	
	J23119 (Const.)	rAPOBEC1	enCas9	P_Xyls_ (Induc.)	Xyls-rAPOBEC1-enCas9-1×UGI	ND	Nontoxic	
	J23119 (Const.)	rAPOBEC1 (W90Y, R126E)	enCas9	ParaBAD (Induc.)	ParaBAD-rAPOBEC1(W90Y, R126E)-enCas9-1×UGI	4 to 7	Nontoxic	
*Bacillus subtilis*	P_veg_ (Const.)	-	nCas9	P_grac_ (Induc.) **	P_grac_-nCas9	-	Toxic	[11]
	P_veg_ (Const.)	PmCDA1	nCas9	P_grac_ (Induc.) **	P_grac_-nCas9-PmCDA1	-	Toxic	
	P_veg_ (Const.)	PmCDA1	dCas9	P_grac_ (Induc.) **	P_grac_-dCas9-PmCDA1	1 to 5	Nontoxic	
*Streptomyces* spp.	J23119 (Const.)	PmCDA1	dCas9	PermE* (Const.)	ermEp*-dCas9-PmCDA1-1×UGI-LVA.Tag	-	Toxic	[13]
	J23119 (Const.)	PmCDA1	dCas9	PtipA (Induc.) **	PtipA-dCas9-PmCDA1-1×UGI-LVA.Tag	1 to 5	Nontoxic	
*Bacillus subtilis*	araABCD (Induc.)	-	nCas9	P_grac_ (Induc.) **	P_grac_-nCas9	-	Toxic	[12]
	araABCD (Induc.)	PmCDA1	dCas9	P_grac_ (Induc.) **	P_grac_-dCas9-PmCDA1-1×UGI-LVA.Tag	1 to 5	Nontoxic	
	araABCD (Induc.)	rAPOBEC1	dCas9	P_grac_ (Induc.) **	P_grac_-rAPOBEC1-dCas9-PmCDA1-1×UGI-LVA.Tag	ND	Nontoxic	
*Paenibacillus polymyxa* E681	araABCD (Induc.)	PmCDA1	dCas9	P_spac_ (Induc.) **	P_spac_-dCas9-PmCDA1-1×UGI-LVA.Tag	ND	Nontoxic	
*Agrobacterium* spp.	J23119 (Const.)	PmCDA1	dCas9	PaadA (Const.)	PaadA-dCas9-PmCDA1-1×UGI-LVA.Tag		Toxic	[14]
	J23119 (Const.)	PmCDA1	dCas9	PvirB (Induc.) **	PvirB-dCas9-PmCDA1-1×UGI-LVA.Tag	ND	Nontoxic	

* enCas9, enhanced Cas9 (K848A/K1003A/R1060A); ** leaky expression by inducible promoter was sufficient for efficient base editing; Const., constitutive promoter; Induc., inducible promoter; UGI, uracil DNA glycosylase inhibitor; ND, no data or analysis done about editing window; # editing window varies with gRNA length.

## Data Availability

All datasets supporting the conclusions of this article are included in the article and Appendix A.

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
