# Peer review of "In Vivo Rapid Investigation of CRISPR-Based Base Editing Components in Escherichia coli (IRI-CCE): A Platform for Evaluating Base Editing Tools and Their Components"

_ijms, 2022, doi:10.3390/ijms23031145_

Round 1

Reviewer 1 Report

This manuscript details the use of base editing in E. coli. The ms provides a nice overview of the current reports of BE in bacteria and outlines the problems with toxicity. Using different strength promoters to drive BE and gRNA expression, the report identifies promoters that allow for cell survival and high levels of editing activity. There is then a comparison of different deaminases, targets and spacer lengths. In my own group’s work, we have actually observed BE in E. coli using similar plant promoters and AtU6, so I have no doubt of the activity reported here. Overall the text is clear, the presentation of the data very approachable, and I find the M&M sufficient for reproduction.

Below, please find a few major and minor comments.

Major:

Several places state that the results obtained here can be useful for pre-testing in plants or other eukaryotes. Should be careful. It is true that some sort of pre-screening step would be advantageous before using a gRNA in stable plant transformation. I am a bit skeptical of using IRI-CCE for these purposes though for a number of reasons. 1, plant protoplasts are routinely used to screen components and can do so within 1-2 days using very similar methods and results. The cloning is also easier as the gRNA target is already present in the genome. 2, cellular difference between bacteria and eukaryotes it pretty big. E. coli can have different BE repair profiles (e.g. Zhao et al 2020). 3, at least when using base editing, one often does not have many options when selecting a gRNA to make the edit. Either that single gRNA will work or you cannot make the desired change. So screening doesn’t really help. So I think the authors need to tone down the pre-testing aspect of IRI-CCE or present some clear examples how this is easier than say a protoplast based screen.

The authors could suggest IRI-CCE could be useful for species without readily available protoplast systems. I think IRI-CCE could also be useful for making novel BE architectures to see qualitatively if they work or not. IRI-CCE may be more sensitive than in planta methods (though this is untested).

I also have some general questions. It would be nice to include the answers in the results. Was there any editing outside the windows for ABEs as detailed for the CBEs? Were other types of repair outcomes observed for the CBEs? E.g. C to G or A?

Minor

Not sure what line 17-19 refers to.

Line 73-74. What is the consensus sequence of AtU6?

Line 243: specify which BE tools have not been reported in E. coli.

Line 245: specify which one, or both, if they do not have a sequence context pref. I thought all deaminases had some preferences. E.g. Cheng et al. xxxx

Line 267: statement about middle positions not being available to A3A is not well supported. It is based on 1 gRNA target (3). The others lack Cs in this position.

Line 269-270: not sure what the UGI sentence is trying to say. Found it confusing.

Line 272: I don’t think it can be said that the CBEs are non-toxic. The CBEs could be killing a lot of the cells, they are just not being recovered on the plates. Some clearly survive, and there appears to be no difference between the different CBEs, but the transformation experiment needs a non-BE control to compare to.

Line 309: protein levels are not quantified, so cannot say high or low amounts of proteins are responsible for the different levels of editing.

Please include in figure legends when EditR is used for quantification. Or include at first use in the results.

Figure 9, why is there a vertical yellow bar at position -5? Please explain in the legend.

Author Response

This manuscript details the use of base editing in E. coli. The ms provides a nice overview of the current reports of BE in bacteria and outlines the problems with toxicity. Using different strength promoters to drive BE and gRNA expression, the report identifies promoters that allow for cell survival and high levels of editing activity. There is then a comparison of different deaminases, targets and spacer lengths. In my own group’s work, we have actually observed BE in E. coli using similar plant promoters and AtU6, so I have no doubt of the activity reported here. Overall, the text is clear, the presentation of the data very approachable, and I find the M&M sufficient for reproduction.

Response: Thank you so much for your positive evaluation and valuable comments.

Below, please find a few major and minor comments.

Major:

Point 1: Several places state that the results obtained here can be useful for pre-testing in plants or other eukaryotes. Should be careful. It is true that some sort of pre-screening step would be advantageous before using a gRNA in stable plant transformation. I am a bit skeptical of using IRI-CCE for these purposes though for a number of reasons. 1, plant protoplasts are routinely used to screen components and can do so within 1-2 days using very similar methods and results. The cloning is also easier as the gRNA target is already present in the genome. 2, cellular difference between bacteria and eukaryotes it pretty big. E. coli can have different BE repair profiles (e.g. Zhao et al 2020). 3, at least when using base editing, one often does not have many options when selecting a gRNA to make the edit. Either that single gRNA will work or you cannot make the desired change. So screening doesn’t really help. So I think the authors need to tone down the pre-testing aspect of IRI-CCE or present some clear examples how this is easier than say a protoplast based screen.

The authors could suggest IRI-CCE could be useful for species without readily available protoplast systems. I think IRI-CCE could also be useful for making novel BE architectures to see qualitatively if they work or not. IRI-CCE may be more sensitive than in planta methods (though this is untested).

Response: Thank you very much for your critical assessment and comment to improvise the manuscript. We agree that the pre-testing of CRISPR reagents in the IRI-CCE platform may provide limited information. Pre-existing methods like protoplast and agroinfiltration-based assays can be used in the case of some model plant species. We also agree with Reviewer 1 comment about using a protoplast system for the same purpose. Protoplast system is possible only in a limited number of plant species, which is also not simple to be established in every lab. It needs expensive sequencing services like deep DNA sequencing to evaluate the experimental outcomes (Lin et al., 2018; 10.1111/pbi.12870). Although with limited information, IRI-CCE is simple, easy to design, and Sanger sequencing can be used for genotyping. IRI-CCE may complement other methods for pre-screening of CRISPR reagents. We have added this aspect in the revised draft, toning down the text, and included the current bottlenecks of IRI-CCE for pre-testing of BE reagents.

In addition, we included the point suggested by Reviewer 1 (and Reviewer 3) about the possible use of IRI-CCE for species without readily available protoplast systems or testing novel BE architectures in the result and discussion.

Result section-

Line 376-379

… Generally, protoplast and transient agroinfiltration assays are used for preliminary screening in plants, with some limitations elaborated in the discussion section. In that scenario, the IRI-CCE platform may complement these existing platforms allowing the simultaneous evaluation of designed bioparts and functionality of the selected sgRNAs….

Line 423-425

… Especially, IRI-CCE may be more sensitive than in planta methods and could be helpful in screening gRNAs for plant species without readily available protoplast or agroinfiltration systems.

Discussion section-

Line 490-491

In addition, IRI-CCE could also be useful for testing novel BE architectures.

Line 579-598

The IRI-CCE system in its present form has some limitations about validating plant-related factors while optimizing BE modules or gRNA screening. Some plant-related factors cannot be assessed in this bacteria-based IRI-CCE system that includes the effect of genome organization on target accessibility, the feature to evaluate nuclear localization signals (NLS) function, different DNA repair machinery in bacteria, and difference in growth temperature of bacteria and plants. However, IRI-CCE offers an efficient way for bacterial BE studies, evidenced by its successful use for optimizing nCas9-based different BE versions (PmCDA1, evoCDA1, APOBEC3A, ABE8e, and ABE9e), which is also the first report in E. coli to the best of our knowledge. In addition, IRI-CCE allows verification of the sgRNA-related factors like nCas9-BE-sgRNA complex formation and target DNA-binding irrespective of the host system. Some existing platforms can be helpful for pre-screening of gRNAs and preliminary check of CRISPR-BE outcomes. For instance, the combination of protoplast culture and pre-assembled ribonucleoprotein (RNP) complexes is one of the handy methods for assessing the editing activities of sgRNAs in plants [51].

Nevertheless, protoplast-based assays require protoplast isolation limiting to only model plant species and purified Cas-sgRNA RNP complexes, which further adds to the cost. Transient agroinfiltration assay proved to be a comparatively quick in vivo method for analyzing sgRNA efficacy, but the lower activity and inconsistency depending on target plant species are the significant obstacles [52]. The IRI-CCE platform may serve as a fast way to validate some key aspects. For instance, ….

Point 2: I also have some general questions. It would be nice to include the answers in the results. Was there any editing outside the windows for ABEs as detailed for the CBEs? Were other types of repair outcomes observed for the CBEs? E.g. C to G or A?

Response: Thank you for the suggestion to include the critical points in the result section. In the case of ABEs, editing activity outside the canonical editing window was not observed. Some studies reported C-to-G or C-to-A as CBEs editing byproducts; we did not observe this phenomenon in the case of improved CBE versions. We have added these points in the revised manuscript in appropriate sections.

Line 359-360

In the case of ABEs, editing activity outside the gRNA window was not observed.

Line 567-569

Some studies reported C-to-G or C-to-A as CBEs editing byproducts [5,8]; we did not observe this phenomenon in the case of basic (PmCDA1, APOBEC3A) or improved CBE (evoCDA1) versions, which might be too low to detect with Sanger sequencing if any.

Point 3: Minor

Not sure what line 17-19 refers to.

Response: Thank you. A sentence is revised for clarity.

Line 18-20

Improved variants of two main BEs, cytidine base editor (CBE) and adenine base editor (ABE), capable of converting C-to-T and A-to-G, respectively, have been recently developed but yet to be tested for editing characteristics in bacteria.

Line 73-74. What is the consensus sequence of AtU6?

Response: The AtU6 promoter consists of two conserved elements: the upstream sequence element (USE; consensus sequence RTCCCACATCG) and a TATA-like box (consensus sequence TTTATATA). This aspect is highlighted in revised Figure 1 (Line 166-169).

Line 243: specify which BE tools have not been reported in E. coli.

Response: Thank you. This is the first study investigating BE variants, namely evoCDA1, APOBEC3A, ABE8e, and ABE9e for E. coli use. We included this information in line 275-276.

Line 245: specify which one, or both, if they do not have a sequence context pref. I thought all deaminases had some preferences. E.g. Cheng et al. xxxx

Response: Thank you. We re-checked the relevant references to check the sequence context preferences for evoCDA1 and APOBEC3A. We are here providing the snaps from a paper from the David Liu group (Huang et al., 2021), which states that evoCDA1 has no context preferences and APOBEC3A has only moderate TC preference. We revised the text for clarity in line 279-281.

Line 267: statement about middle positions not being available to A3A is not well supported. It is based on 1 gRNA target (3). The others lack Cs in this position.

Response: Thank you for pointing this aspect. We agree with the Reviewer that middle position Cs (from 12-16) can be accessed by A3A-mediated deamination in other target sites. Data interpretation for positions 12 to 16 is made based on testing of only one sgRNA (gRNA target 3). We have revised the sentence for clarity.

Line 301-304

The upstream (positions -3 to -1) regions of target sites were not accessible to A3A deamination in the tested combinations of sgRNAs and respective target sites (Figure 6C). In addition, A3A-mediated C-to-T editing at middle positions (12 to 16) in Test sgRNA3 was not observed.

Line 269-270: not sure what the UGI sentence is trying to say. Found it confusing.

Response: We removed the UGI sentence. Thank you.

Line 272: I don’t think it can be said that the CBEs are non-toxic. The CBEs could be killing a lot of the cells, they are just not being recovered on the plates. Some clearly survive, and there appears to be no difference between the different CBEs, but the transformation experiment needs a non-BE control to compare to.

Response: Thank you for the suggestion. We agree that CBEs could kill some cells and decreases in cell survival rate compared to non-BE as a control. For clarity, we revised the text.

Line 303-306

The comparison between nCas9(D10A) and the CBE types (PmCDA1, evoCDA1, and A3A) containing rppH sgRNA3 targeting rppH gene indicated that all CBEs are less toxic with slight variation between pGlpT and p35S(L)I promoter tests (Figure S4) and survived cells allowing BE activity studies.

Line 309: protein levels are not quantified, so cannot say high or low amounts of proteins are responsible for the different levels of editing.

Response: The sentence is revised. Thank you.

Line 360-362

Overall, all CBEs and ABEs data indicate that the differential-strength promoters may contribute to the variable editing outcomes within the canonical editing window.

Please include in figure legends when EditR is used for quantification. Or include at first use in the results.

Response: Thank you for your suggestion. As you suggested, we added EditR in the caption of Figure 3.

Figure 9, why is there a vertical yellow bar at position -5? Please explain in the legend.

Response: Thank you. Information of yellow bar included in the legend of Figure 9 (in the revised draft- Figure 10).

Reviewer 2 Report

The manuscript described by Shelake et al. provided a platform technology for evaluating CRISPR-based DAB-free base editing tools and their components using E. coli cells. The method is potentially very useful for analyzing and optimizing the CRISPR base editors. The manuscript is well-written and overall understandable for researchers in the field. The following minor points should be addressed for revision.

The abbreviation in the title (IRI-CCE) should be spelled-out.

The author who has the second affiliation is missing.

Author Response

The manuscript described by Shelake et al. provided a platform technology for evaluating CRISPR-based DAB-free base editing tools and their components using E. coli cells. The method is potentially very useful for analyzing and optimizing the CRISPR base editors. The manuscript is well-written and overall understandable for researchers in the field. The following minor points should be addressed for revision.

Response: We thank Reviewer 2 for finding our manuscript of interest and useful to the readers.

Point 1: The abbreviation in the title (IRI-CCE) should be spelled out.

Response: Thank you for suggesting the revision in the title. We agree with the Reviewer 2 and IRI-CCE is spelled out in the revised draft.

In vivo Rapid Investigation of CRISPR-based Base Editing Components in Escherichia coli (IRI-CCE): A Platform for Evaluating Base Editing Tools and Its Components

Point 2: The author who has the second affiliation is missing.

Response: Authors are thankful for pointing an error in drafting the manuscript in IJMS format. The second affiliation is included at the appropriate place in the Author list.

Reviewer 3 Report

In this research article, Shelake & al assessed different base editors activity in E.coli by targeting a reporter plasmid or endogenous sites. They show that assessing base editing activity is facilitated by using promoters with lower strength. They claim that their findings could be used as a platform for rapid assessment and optimization of base editors activity.

Although the authors detect base editing for multiple BEs at high efficiency, I do not think that this study provides sufficient evidences for reduced toxicity of their system compared to existing ones. Furthermore, I do not think that they convincingly demonstrate that their system can be used as a platform for optimizing base editors as they exclusively tested architectures that were previously validated in other organisms.

Major comments:

1) The authors detected toxicity when using pEc1:nCas9-CBE in combination with pJ23119:gRNA. It is not clear whether toxicity is caused by the gRNA or the BE. Furthermore, other studies have used dCas9 instead of nCas9 together with the pJ23119, or CBEs with or without UGI.  It would be important that the authors systematically compare all combinations to clearly identify the cause of the toxicity and whether indeed the choice of promoters (driving BE or gRNA) is the key parameter.

The authors also state that bacterial strains does not have impact on toxicity but do not show the quantifications for this claim.

2) It is not clear to me how applicable can this platform be for optimizing base editors and transferring this knowledge to other species (including plants). Although the system is quick and require only few cloning steps, I think that the base editors tested and the limited number of parameters analyzed could have been done directly in plant protoplasts or animal cells (as is done in multiple published reports). In practice gRNA design is not a limiting factor when testing BEs. The authors do not clearly demonstrate how their system could be leveraged to optimize base editors but rather only test few architectures known to be functional in animal or plant. Bacterial system presents several limitations for optimizing BE (cannot assess NLS function, genome size difference, editing the plasmid that carries the BE and gRNA, temperature) and thus the authors should demonstrate how they want to use the bacterial system for optimizing BEs.

In the description of their system, there is no data on the use of the GFP fluorescence to detect in frame gRNA clones. Also they discuss but do not show data on one vs two plasmid components assays when assessing BE activity.

Minor comments:

- I think that the data on different editing window is over-interpreted. The difference are very mild on all target sites for ABE or CBE and likely the result of different BE activity combined with lower detection resolution when using Sanger sequencing (<15% efficiency not reliably detected). In Fig8 and Fig9 it would be important to show error bars to better appreciate the variability between samples.

- Does dCas9-BE do show lower efficiency than nCas9-BE. Why not using dCas9 to test other elements of BE if the construct is less toxic?

Author Response

In this research article, Shelake & al assessed different base editors activity in E. coli by targeting a reporter plasmid or endogenous sites. They show that assessing base editing activity is facilitated by using promoters with lower strength. They claim that their findings could be used as a platform for rapid assessment and optimization of base editors activity.

Although the authors detect base editing for multiple BEs at high efficiency, I do not think that this study provides sufficient evidences for reduced toxicity of their system compared to existing ones. Furthermore, I do not think that they convincingly demonstrate that their system can be used as a platform for optimizing base editors as they exclusively tested architectures that were previously validated in other organisms.

Response: We sincerely appreciate all constructive comments and suggestions, which helped us to improve the quality of our article.

Major comments:

Point 1: 1) The authors detected toxicity when using pEc1:nCas9-CBE in combination with pJ23119:gRNA. It is not clear whether toxicity is caused by the gRNA or the BE.

Response: Thank you for commenting on this point. Toxicity was observed only in the combination BE and sgRNA expression, not when only BE or sgRNA was independently expressed. This aspect is summarized in Supplementary Table S1.

Point 2: Furthermore, other studies have used dCas9 instead of nCas9 together with the pJ23119, or CBEs with or without UGI.  It would be important that the authors systematically compare all combinations to clearly identify the cause of the toxicity and whether indeed the choice of promoters (driving BE or gRNA) is the key parameter.

Response: Thank you so much for your excellent comment. We agree that the information about cytotoxicity was not adequately summarized in the earlier version and are very sorry for this negligence. The tested combination of promoters expressing BE components and their effect on cytotoxicity in E. coli are provided in newly added Supplementary Table S1, which may guide readers to follow the content. This aspect is also discussed with proper citations in the second paragraph of the discussion section.

We agree with Reviewer 3 that the cytotoxicity caused by BE components is a well-known issue in prokaryotic BE experiments (already summarized in Table 1). Although the molecular mechanism associated with dCas9 binding to genomic regions is still elusive, higher expression of dCas9 only (without sgRNA) binds genomic regions and damages E. coli cells, as reported in some papers (Cho et al., 2018; ACS Synth Biol.) (Zhang and Voigt., 2018; NAR). In the case of BE, the expression of dCas9-PmCDA1-1xUGI was failed in the earlier study, possibly because excess UGI promoted mutagenesis and compromised genome integrity in E. coli (Banno et al., 2018). Possible reasons underlying the toxicity of BE reagents in bacteria would be complicated, and detailed mechanisms remained to be uncovered. In the end, it is tricky to know how high expression of BE machinery could be optimal for cell survival and BE test in E. coli.

A protein degradation LVA tag decreases the half-life of the fusion protein and reduces the toxicity to an acceptable level. The expression of dCas9-PmCDA1-UGI-LVA was successful for BE studies in E. coli (Banno et al., 2018).

The ultimate goal of our work was to establish a BE testing platform with the nCas9 (and without the fusion of LVA tag) because it is preferred in other organisms. We opted to use a combination of bacteria/plant promoters to achieve the expression of nCas9-based BE components with a sufficient cell survival rate allowing BE testing and direct use of modular clones for further studies in plants. Some of the modular bioparts (pAtU6, p35S(L)I) adopted for bacterial use in current work are successfully applied in plants, as evident in our published papers (Pramanik et al., 2021), and experiments are underway to establish optimized BE tools in plants, especially in tomato. We will also compare BE efficacy in our IRI-CCE and previous approaches in the near future, which is not the focus of current work.

Point 3: The authors also state that bacterial strains does not have impact on toxicity but do not show the quantifications for this claim.

Response: Thank you. The data about cytotoxicity and CBE activities in four strains with Target-AID constructs are included in the revised draft as Supplementary Figure S6 and Figure 5, respectively. We agree with Reviewer 3 that variable cell survival rates are possible in different strains. We also observed a slight variation in cell survival between different strains. Therefore, newly added data about the CBE test affirms the statement made in the article, i.e., Irrespective of the genetic background of strain, we found comparable C-to-T mutations in tested sgRNAs in all three bacterial strains (Line 265-269).

Point 4: 2) It is not clear to me how applicable can this platform be for optimizing base editors and transferring this knowledge to other species (including plants). Although the system is quick and require only few cloning steps, I think that the base editors tested and the limited number of parameters analyzed could have been done directly in plant protoplasts or animal cells (as is done in multiple published reports). In practice gRNA design is not a limiting factor when testing BEs. The authors do not clearly demonstrate how their system could be leveraged to optimize base editors but rather only test few architectures known to be functional in animal or plant. Bacterial system presents several limitations for optimizing BE (cannot assess NLS function, genome size difference, editing the plasmid that carries the BE and gRNA, temperature) and thus the authors should demonstrate how they want to use the bacterial system for optimizing BEs.

Response: Thank you. The novelty of our work was not well reflected in the writing of the previously submitted draft. We have modified our manuscript following the suggestions of Reviewers. We hope that the revised version will effectively deliver the key findings of this work that include-

  1. We find the promoter-terminator combinations that could control the expression of nCas9-based BE and sgRNA to allow cell survival in E. coli., eventually leading to successful base editing.
  2. To the best of our knowledge, this is the first report showing the exogeneous pAtU6 promoter-mediated sgRNA expression is efficient for BE studies in E. coli.
  3. This is the first study reporting characterization of nCas9-based improved variants for E. coli., especially evoCDA1, APOBEC3A, ABE8e, and ABE9e.
  4. The evaluation of modular parts for various characteristics in the IRI-CCE platform provides a significant advantage for further investigations to avoid loss of time.
  5. We show that the editing window length is adjustable using the extended or truncated gRNAs and the differential-strength promoters.

The findings reported in the current work provides an optimized set of promoter-terminator combination, biopart modules, and validation strategies of newer BE reagents using the IRI-CCE platform.

We agree with Reviewer 3 that the previous studies have circumvented the toxicity issues by using different ways like replacing nCas9 with dCas9 combined with the use of LVA tag in E. coli (Banno et al., 2018). We also agree that our IRI-CCE system has some limitations. We would like to point out that none of the previous studies has succeeded the nCas9-based BE test in E. coli which is primarily used for other organisms. E. coli is a preferred host for gene cloning, and hence use of nCas9 as evidently used in IRI-CCE could be helpful in several ways. The following sentences are added in the revised manuscript for clarity to address the comments of Reviewers 1 and 3.

Result section-

Line 376-379

… Generally, protoplast and transient agroinfiltration assays are used for preliminary screening in plants, with some limitations elaborated in the discussion section. In that scenario, the IRI-CCE platform may complement these existing platforms allowing the simultaneous evaluation of designed bioparts and functionality of the selected sgRNAs….

Line 423-425

… Especially, IRI-CCE may be more sensitive than in planta methods and could be helpful in screening gRNAs for plant species without readily available protoplast or agroinfiltration systems.

Discussion section-

Line 490-491

In addition, IRI-CCE could also be useful for testing novel BE architectures.

Line 579-598

The IRI-CCE system in its present form has some limitations about validating plant-related factors while optimizing BE modules or gRNA screening. Some plant-related factors cannot be assessed in this bacteria-based IRI-CCE system that includes the effect of genome organization on target accessibility, the feature to evaluate nuclear localization signal (NLS) function, different DNA repair machinery in bacteria, and difference in growth temperature of bacteria and plants. However, IRI-CCE offers an efficient way for bacterial BE studies, evidenced by its successful use for optimizing nCas9-based different BE versions (PmCDA1, evoCDA1, APOBEC3A, ABE8e, and ABE9e), which is also the first report in E. coli to the best of our knowledge. In addition, IRI-CCE allows verification of the sgRNA-related factors like nCas9-BE-sgRNA complex formation and target DNA-binding irrespective of the host system. Some existing platforms can be helpful for pre-screening of gRNAs and preliminary check of CRISPR-BE outcomes. For instance, the combination of protoplast culture and pre-assembled ribonucleoprotein (RNP) complexes is one of the handy methods for assessing the editing activities of sgRNAs in plants [51].

Nevertheless, protoplast-based assays require protoplast isolation limiting to only model plant species and purified Cas-sgRNA RNP complexes, which further adds to the cost. Transient agroinfiltration assay proved to be a comparatively quick in vivo method for analyzing sgRNA efficacy, but the lower activity and inconsistency depending on target plant species are the significant obstacles [52]. The IRI-CCE platform may serve as a fast way to validate some key aspects. For instance, ….

Point 5: In the description of their system, there is no data on the use of the GFP fluorescence to detect in frame gRNA clones.

Response: Thank you for your comment. We did not use the sfGFP fluorescence to detect in-frame gRNA clones. Instead, we used in-frame sfGFP as a reporter for successful cloning of desired target DNA sequences in the downstream position in the universal target-acceptor module. The sentence is revised for clarity.

Line 181-182

Cloning of the required target DNA region into a sfGFP-based universal target-acceptor using MoClo kit protocol [17] permits the assessment of any intended sgRNA.

Point 6: Also they discuss but do not show data on one vs two plasmid components assays when assessing BE activity.

Response: No significant difference was observed in the single and two-plasmid assay; therefore, it was not focused further in work. We have included the data of the two-plasmid system as Supplementary Figure S4.

Point 7: Minor comments:

- I think that the data on different editing window is over-interpreted. The difference are very mild on all target sites for ABE or CBE and likely the result of different BE activity combined with lower detection resolution when using Sanger sequencing (<15% efficiency not reliably detected). In Fig8 and Fig9 it would be important to show error bars to better appreciate the variability between samples.

Response: Thank you. We agree with the reviewer that the EditR program is not so accurate as deep sequencing for base conversions, but it is rapid, straightforward, and inexpensive. For many cases, especially when comparing different base editors in the same target sites, it is good enough because it can detect as small as 2.5% difference (Kluesner et al., 2018). Therefore, many labs used this method in their publications. BE interpretation with EditR is mainly limited by the quality of Sanger sequencing data. Perhaps the isolation of target-containing plasmids facilitates an easier way to prepare high-quality DNA templates for Sanger sequencing.

As suggested by Reviewer 3, we have revised figures 8 and 9 (Figure 9 and 10 in the revised manuscript) and added the error bars to reflect variability between samples.

- Does dCas9-BE do show lower efficiency than nCas9-BE. Why not using dCas9 to test other elements of BE if the construct is less toxic?

Response: Thank you. As mentioned in response to points 1 and 2, optimizing a system with nCas9 was our primary objective because we wanted to use designed modules ultimately in plants. We certainly agree with Reviewer 3’s opinion that dCas9 may be less toxic for cell growth. We have included the data with dCas9-CBE in Supplementary figure S5. In the future, we will surely consider this aspect to investigate the dCas9 effect on the editing efficiency of different BEs.
